# Learning Graph Representations by Dendrograms

## Abstract

Hierarchical clustering is a common approach to analysing the multi-scale structure of graphs observed in practice. We propose a novel metric for assessing the quality of a hierarchical clustering. This metric reflects the ability to reconstruct the graph from the dendrogram encoding the hierarchy. The best representation of the graph for this metric in turn yields a novel hierarchical clustering algorithm. Experiments on both real and synthetic data illustrate the efficiency of the approach.

## 1 Introduction

Many datasets have a graph structure. Examples include infrastructure networks, communication networks, social networks, databases and co-occurence networks, to quote a few. These graphs often exhibit a complex, multi-scale structure where each node belong to many groups of nodes, so-called clusters, of different sizes (Caldarelli and Vespignani (2007)).

Hierarchical graph clustering is a common technique for the analysis of the multi-scale structure of large graphs. Instead of looking for a single partition of the set of nodes, as in usual clustering techniques, the graph is represented by a hierarchical structure known as a dendrogram, which can then be used to find relevant clusterings at different resolutions, by suitable cuts of this dendrogram.

While many hierarchical graph clustering algorithms have recently been proposed (see for instance Newman (2004); Pons and Latapy (2005); Sales-Pardo et al. (2007); Clauset et al. (2008); Lancichinetti et al. (2009); Ronhovde and Nussinov (2009); Huang et al. (2010); Chang et al. (2011); Tremblay and Borgnat (2014); Bonald et al. (2018)), it proves very difficult to evaluate their performance due to the absence of public datasets of graphs with ground-truth hierarchy. A natural approach is then to define some quality metric based on the graph itself, just like modularity is a popular metric for assessing the quality of graph clustering (Newman and Girvan (2004)).

A cost function has been proposed by Dasgupta (2016) for hierachical graph clustering and has been further analysed and extended by Cohen-Addad et al. (2018); it can be viewed as the expected size of the smallest cluster induced by the hierarchy and containing two nodes sampled at random from the edges of the graph. In this paper, we propose a quality metric based on the ability to reconstruct the graph from the hierarchy. It is equal to the relative entropy of the probability distribution on node sets induced by the hierarchy compared to that induced by independent node sampling. Finding the best graph representation for this metric yields a novel hierarchical clustering algorithm, which can in turn be interpreted in terms of modularity.

In the next section, we introduce sampling distributions of nodes and node sets induced by the graph, which play a key role in our approach. We then formalize the problem of graph representation by a dendrogram. Our quality metric follows from the characterization of the optimal solution in terms of graph reconstruction. The corresponding hierarchical graph clustering algorithm is then presented and interpreted in terms of modularity. The results are illustrated by some experiments on both real and synthetic data.

## 2 SAMPLING DISTRIBUTION

Consider a weighted, undirected, connected graph $G = (V, E)$ of $n$ nodes, without self-loops. Let $w(u, v)$ be equal to the weight of edge $\{u, v\}$, if any, and to 0 otherwise. We refer to the weight of node $u$ as:

$$w(u) = \sum_{v \in V} w(u, v).$$

We denote by $w$ the total weight of nodes:

$$w = \sum_{u \in V} w(u) = \sum_{u,v \in V} w(u, v).$$

Observe that the weight of each edge is counted twice in this sum.

Similarly, for any sets $A, B \subset V$, let

$$w(A, B) = \sum_{u \in A, v \in B} w(u, v),$$

and

$$w(A) = \sum_{u \in A} w(u).$$

**Node sampling.** The weights induce a probability distribution on node pairs:

$$\forall u, v \in V, \quad p(u, v) = \frac{w(u, v)}{w},$$

with marginal distribution:

$$\forall u \in V, \quad p(u) = \sum_{v \in V} p(u, v) = \frac{w(u)}{w}.$$

This joint probability distribution is symmetric in the sense that $p(u, v) = p(v, u)$ for all $u, v \in V$. This is the relative frequency of moves from node $u$ to node $v$ by a random walk in the graph, with transition probability:

$$\forall u, v \in V, \quad p(v|u) = \frac{p(u, v)}{p(u)} = \frac{w(u, v)}{w(u)}.$$

Observe that $p(u, u) = 0$ for all $u \in V$ due to the absence of self-loops in the graph.

**Node set sampling.** For any partition $\mathcal{P}$ of $V$, the weights induce a probability distribution on $\mathcal{P}$:

$$\forall A, B \in \mathcal{P}, \quad p(A, B) = \frac{w(A, B)}{w},$$

with marginal distribution:

$$\forall A \in \mathcal{P}, \quad p(A) = \sum_{B \in \mathcal{P}} p(A, B) = \frac{w(A)}{w}.$$

Again, this joint probability distribution is symmetric in the sense that $p(A, B) = p(B, A)$ for all $A, B \in \mathcal{P}$. This is the relative frequency of moves from $A$ to $B$ by the random walk.

## 3 REPRESENTATION BY A DENDROGRAM

Assume the graph $G$ is represented by a dendrogram, that is a rooted binary tree $\mathcal{T}$ whose leaves are the nodes of the graph, $V$. We denote by $\text{int}(\mathcal{T})$ the set of internal nodes of the tree $\mathcal{T}$ (all nodes except leaves). For each $i \in \text{int}(\mathcal{T})$, a positive number $d(i)$ is assigned to node $i$, corresponding to its height in the dendrogram. We assume that the dendrogram is regular in the sense that $d(i) \geq d(j)$ if $i$ is an ancestor of $j$ in the tree. For each $i \in \text{int}(\mathcal{T})$, there are two subtrees attached to node $i$, with sets of leaves $A$ and $B$; these sets uniquely identify the internal node $i$ so that we can write $i = (A, B) = (B, A)$ and $d(i) = d(A, B) = d(B, A)$.

**Ultrametric.** The dendrogram defines a metric on $V$: for each $u, v \in V$, we define $d(u, v) = 0$ if $u = v$ and $d(u, v) = d(i)$ otherwise, where $i \in \text{int}(\mathcal{T})$ is the closest common ancestor of $u$ and $v$ in the corresponding tree. This is an ultrametric in the sense that:
$$\forall u, v, x \in V, \quad d(u, v) \leq \max(d(u, x), d(v, x)).$$
Conversely, any ultrametric defines a dendrogram, which can be built from bottom to top by successive merges of the closest nodes, where the distance of the node resulting from the merge of two nodes $u$ and $v$, which we denote by $u \cup v$, to any other node $x$ is defined by:
$$d(u \cup v, x) = \max(d(u, x), d(v, x)).$$

**Nested clustering.** The tree $\mathcal{T}$ induces a clustering $\mathcal{C}$ of the set of nodes, with $A \in \mathcal{C}$ if and only if $A$ is the set of leaves of a subtree of $\mathcal{T}$. This is a nested clustering in the sense that:
$$\forall A, B \in \mathcal{C}, \quad A \subset B \quad \text{or} \quad A \cap B = \emptyset.$$
Although $\mathcal{C}$ is not a partition of $V$, we get a probability distribution on cluster pairs by restricting the sampling to cluster pairs $A, B$ such that $(A, B) \in \text{int}(\mathcal{T})$:
$$\sum_{A, B : (A, B) \in \text{int}(\mathcal{T})} p(A, B) = \sum_{i \in \text{int}(\mathcal{T})} \sum_{u, v : i = u \wedge v} p(u, v) = \sum_{u, v \in V} p(u, v) = 1,$$
where $u \wedge v$ denotes the closest common ancestor of $u$ and $v$ in the tree $\mathcal{T}$.

**Graph representation.** To assess the quality of the dendrogram, we address the issue of the reconstruction of the graph from the dendrogram. In general, some information about the relative node weights may be known in addition to the dendrogram. Thus we introduce some probability distribution $\pi$ on $V$ representing this prior information: the distribution $\pi$ is uniform in the absence of such information and is equal to the sampling distribution $p$ for a perfect knowledge of the relative node weights. Now we look for the best representation of $G$ by a dendrogram $d$ in the sense of the reconstruction of $G$ from $d, \pi$, which can be viewed as the autoencoding scheme:
$$G \longmapsto d, \pi \longmapsto \hat{G},$$
where $\hat{G}$ is the reconstruction of $G$.

Since $d(u, v)$ can be interpreted as a distance between $u$ and $v$, its inverse corresponds to a similarity. Accounting for the prior information about the relative weights of $u$ and $v$, we define the weight $\hat{w}(u, v)$ between nodes $u, v \in V$ in the graph $\hat{G}$ as:
$$\hat{w}(u, v) = \frac{\pi(u)\pi(v)}{d(u, v)} 1_{\{u \neq v\}}. \tag{1}$$
Denoting by $\hat{w}$ the total weight,
$$\hat{w} = \sum_{u, v \in V} \hat{w}(u, v), \tag{2}$$
we get the following node pair sampling distribution associated with $\hat{G}$:
$$\hat{p}(u, v) = \frac{\hat{w}(u, v)}{\hat{w}}. \tag{3}$$
The distance between graphs $G$ and $\hat{G}$ can then be assessed through the Kullback-Leibler divergence between the respective sampling distributions,
$$D(p||\hat{p}) = \sum_{u, v \in V} p(u, v) \log \frac{p(u, v)}{\hat{p}(u, v)},$$
which can be written as the relative entropy $D(p||\hat{p}) = H(p, \hat{p}) - H(p)$, where
$$H(p, \hat{p}) = - \sum_{u, v \in V} p(u, v) \log \hat{p}(u, v) \quad \text{and} \quad H(p) = - \sum_{u, v \in V} p(u, v) \log p(u, v)$$
are the cross-entropy between the probability distributions $p$ and $\hat{p}$ and the entropy of the probability distribution $p$, respectively. Observe that $H(p, \hat{p}) \geq H(p)$, with equality if and only if $\hat{p} = p$.

**Optimization problem.** Minimizing the Kullback-Leibler divergence $D(p||\hat{p})$ in $\hat{p}$ is equivalent to minimizing the cross-entropy $H(p,\hat{p})$ in $\hat{p}$. In view of (1) and (3), we have

$$H(p,\hat{p}) = H(p,\pi) + \sum_{u,v \in V} p(u,v) \log d(u,v) + \log(\hat{w}),$$

where $\pi$ here denotes the bivariate probability distribution $\pi(u,v) = \pi(u)\pi(v)$. Thus the problem reduces to minimizing the cost function:

$$J(d) = \sum_{u,v \in V} p(u,v) \log d(u,v) + \log \left( \sum_{u,v \in V} \frac{\pi(u)\pi(v)}{d(u,v)} \right),$$

over all ultrametrics $d$ defined on $V$. Observe that $J(\alpha d) = J(d)$ for any $\alpha > 0$, so that the best ultrametric is defined up to a multiplicative constant. Using the fact that $d(u,v) = d(i)$, where $i = u \wedge v$ is the closest common ancestor of $u$ and $v$ in the tree, we get:

$$J(d) = \sum_{A,B:(A,B) \in \text{int}(\mathcal{T})} p(A,B) \log d(A,B) + \log \left( \sum_{A,B:(A,B) \in \text{int}(\mathcal{T})} \frac{\pi(A)\pi(B)}{d(A,B)} \right), \quad (4)$$

with $\pi(A) = \sum_{u \in A} \pi(u)$. The problem of the best representation of $G$ by a dendrogram $d$ now reduces to the optimization problem:

$$\arg \min_d J(d), \quad (5)$$

over all ultrametrics $d$ on $V$.

## 4 Optimal representation

We seek to solve the optimization problem (5), for some given graph $G$.

**Optimal distances.** We first assume that the binary tree $\mathcal{T}$ of the ultrametric is given and look for the best corresponding distance $d : \text{int}(\mathcal{T}) \to \mathbb{R}_+$. For each $(A,B) \in \text{int}(\mathcal{T})$, we get by the differentiation of (4) in $d(A,B)$,

$$\frac{p(A,B)}{d(A,B)} = \lambda \frac{\pi(A)\pi(B)}{d(A,B)^2},$$

where

$$\lambda = \left( \sum_{A,B:(A,B) \in \text{int}(\mathcal{T})} \frac{\pi(A)\pi(B)}{d(A,B)} \right)^{-1},$$

that is

$$d(A,B) = \lambda \frac{\pi(A)\pi(B)}{p(A,B)}. \quad (6)$$

**Optimal tree.** Replacing $d(A,B)$ by its optimal value (6) in (4), we deduce that the optimization problem (5) reduces to:

$$\arg \max_{\mathcal{T}} \sum_{A,B:(A,B) \in \text{int}(\mathcal{T})} p(A,B) \log \frac{p(A,B)}{\pi(A)\pi(B)}. \quad (7)$$

The dendrogram is then fully determined by (6), for each internal node $(A,B)$ of the tree $\mathcal{T}$. Thus the objective is now to *maximize* the Kullback-Leibler divergence between two probability distributions on the nested clustering induced by the tree $\mathcal{T}$: $p$ (edge sampling) and $\pi$ (independent node sampling). It provides a meaningful quality metric for hierarchical clustering, given by:

$$\sum_{A,B:(A,B) \in \text{int}(\mathcal{T})} p(A,B) \log \frac{p(A,B)}{\pi(A)\pi(B)}. \quad (8)$$

A key difference between our metric (8), we refer to as the *relative entropy* of the hierarchy, and the cost functions proposed in the literature (Dasgupta (2016); Cohen-Addad et al. (2018)) lies in the entropy term:

$$-\sum_{A,B:(A,B)\in \mathrm{int}(\mathcal{T})} p(A,B)\log p(A,B).$$

Removing this term from (8) and inversing the sign yields the cost function:

$$\sum_{A,B:(A,B)\in \mathrm{int}(\mathcal{T})} p(A,B)(\log \pi(A) + \log \pi(B)),$$

to be compared with usual cost functions, of the form:

$$\sum_{A,B:(A,B)\in \mathrm{int}(\mathcal{T})} p(A,B)(\pi(A) + \pi(B)). \tag{9}$$

When $\pi$ is the uniform distribution (no prior information on the node weights), these cost functions become respectively, up to some normalization constant:

$$\sum_{A,B:(A,B)\in \mathrm{int}(\mathcal{T})} p(A,B)(\log |A| + \log |B|) \quad \text{and} \quad \sum_{A,B:(A,B)\in \mathrm{int}(\mathcal{T})} p(A,B)(|A| + |B|).$$

The latter is Dasgupta's cost function, equal to the expected size of the smallest cluster containing two random nodes sampled from $p$.

**Consistency.** Like Dasgupta's cost function and its extensions considered in Cohen-Addad et al. (2018), the objective function (8) is consistent in the sense that, if the original graph has a hierarchical structure, then any solution to the optimization problem (7) is a tree generating the graph:

**Proposition 1** *Assume that the original graph $G$ is generated by some ultrametric $d_G$, in the sense that*

$$\forall u, v \in V, \quad w(u,v) = \frac{\pi(u)\pi(v)}{d_G(u,v)} 1_{\{u \neq v\}}.$$

*Then any tree $\mathcal{T}$ maximizing (8) is a tree induced by the ultrametric $d_G$.*

*Proof.* We have:

$$H(p) \leq \min_{\hat{p}} H(p,\hat{p}) \leq H(p,\pi) + \min_d J(d)$$

$$= H(p,\pi) - \max_{\mathcal{T}} \sum_{A,B:(A,B)\in \mathrm{int}(\mathcal{T})} p(A,B)\log \frac{p(A,B)}{\pi(A)\pi(B)}.$$

Since $J(d_G) = H(p) - H(p,\pi)$, any tree $\mathcal{T}$ maximizing the objective function (8) admits an ultrametric $d$ such that $J(d) = H(p) - H(p,\pi)$. The corresponding sampling distribution $\hat{p}$, defined by (3), is such that $H(p,\hat{p}) = H(p)$. We deduce that $\hat{p} = p$, so that $d = \alpha d_G$ for some constant $\alpha > 0$ and $\mathcal{T}$ is a tree associated with the ultrametric $d_G$. □

**Absence of bias.** Another property of the objective function (8) is the absence of bias: if the graph $G$ is a clique, then all trees $\mathcal{T}$ have the same relative entropy. This property is also satisfied by Dasgupta's cost function and its extensions (Cohen-Addad et al. (2018)).

**Proposition 2** *If the graph $G$ is a clique with unit weights, then all trees $\mathcal{T}$ have the same relative entropy (8).*

*Proof.* Let $\delta = 1/\pi(u)$, for some $u \in V$. Observe that this quantity is independent of $u$. Now for any tree $\mathcal{T}$, we have:

$$\sum_{A,B:(A,B)\in \mathrm{int}(\mathcal{T})} p(A,B)\log \frac{p(A,B)}{\pi(A)\pi(B)} = \sum_{A,B:(A,B)\in \mathrm{int}(\mathcal{T})} p(A,B)\log \frac{\delta^2}{n(n-1)} = \log \frac{\delta^2}{n(n-1)}.$$

□

## 5  Hierarchical clustering

The optimization problem (8) is combinatorial. We believe that it is NP-hard, like the optimization problem associated with Dasgupta's cost function (Dasgupta (2016)). The optimal distances (6) suggest the following greedy algorithm: start from $n$ clusters (one per node) and successively merge the two closest clusters in terms of inter-cluster distance (6). This is a usual agglomerative algorithm with inter-cluster similarity (so-called linkage):

$$\sigma(A, B) = \frac{p(A, B)}{\pi(A)\pi(B)}. \tag{10}$$

The dendrogram is built from bottom to top, with distance $\sigma(A, B)^{-1}$ attached to the internal node $(A, B)$ resulting from the merge of clusters $A, B$. The linkage is reducible, in the sense that:

$$\sigma(A \cup B, C) \leq \max(\sigma(A, C), \sigma(B, C)).$$

This inequality guarantees that the resulting dendrogram is regular (the sequence of distances attached to successive internal nodes is non-decreasing) and that the corresponding distance on $V$ is an ultrametric.

**Linkage.**  For $\pi$ the uniform distribution (no prior information on the node weights), the linkage (10) is proportional to the usual average linkage:

$$\sigma(A, B) \propto \frac{w(A, B)}{|A||B|},$$

corresponding to the density of the cut separating clusters $A$ and $B$. For $\pi$ equal to $p$ (perfect knowledge of the node weights), this is the linkage proposed in Bonald et al. (2018):

$$\sigma(A, B) = \frac{p(A, B)}{p(A)p(B)}.$$

**Modularity.**  The linkage (10) can be interpreted in terms of modularity. The modularity of any partition $\mathcal{P}$ of the set of nodes $V$ is defined by (Newman and Girvan (2004)):

$$Q = \sum_{C \in \mathcal{P}} \sum_{u,v \in C} (p(u, v) - p(u)p(v)).$$

This is the difference between the probabilities that two nodes belong to the same cluster when sampled from the edges and independently from the nodes, in proportion to their weights. The former sampling distribution depends on the graph while the latter depends on the graph through the node weights only. A more general definition of modularity is:

$$Q = \sum_{C \in \mathcal{P}} \sum_{u,v \in C} (p(u, v) - \pi(u)\pi(v)), \tag{11}$$

where $\pi$ is a probability distribution representing the prior knowledge on the node weights. For instance, it may be uniform (no prior information) or equal to $p$ (the usual definition of modularity, where the information on the relative node weights is known)[1].

The problem of modularity maximization generally provides a single clustering. To get clusterings with different granularities, reflecting the multi-scale structure of real graphs, it is common to introduce some positive parameter $\gamma$, called the resolution, that controls the respective weights of both terms in the definition of modularity (Reichardt and Bornholdt (2006); Lambiotte et al. (2014); Newman (2016)). The modularity of partition $\mathcal{P}$ at resolution $\gamma$ is defined by:

$$Q_\gamma = \sum_{C \in \mathcal{P}} \sum_{u,v \in C} (p(u, v) - \gamma\pi(u)\pi(v)).$$

---

[1]Another common interpretation of modularity is the difference between the proportions of edge weights within clusters in the original graph and in some null model where nodes $u$ and $v$ are linked with probability $\pi(u)\pi(v)$; for $\pi$ the uniform distribution (no prior information on the node weights), the null model is an Erdős-Rényie graph while for $\pi = p$ (perfect knowledge of the node weights), the null model is the configuration model (Van Der Hofstad (2017)).

When $\gamma \to 0$, the second term becomes negligible and the best partition $\mathcal{P}$ is trivial, with a single cluster equal to the set of nodes $V$. When $\gamma \to +\infty$, the second term becomes preponderant and the best partition $\mathcal{P}$ has $n$ clusters, one per node. Now the maximum resolution beyond which the best partition $\mathcal{P}$ has $n$ clusters is given by:

$$\max_{u \neq v} \frac{p(u,v)}{\pi(u)\pi(v)}.$$

The first node pair to be merged (at maximum resolution) is that achieving this maximum, which is the closest pair in terms of linkage (10). The agglomerative algorithm based on linkage (10) can thus be interpreted as the greedy maximization of modularity at maximum resolution.

## 6 EXPERIMENTS

In this section, we show how our metric (8) and Dasgupta's cost function (9) behave on both real and synthetic data. We do not claim that one metric is better than the other and thus the experimental results are presented for illustrative purposes only. We have coded the metrics in Python and grouped the experiments into a Jupyter notebook for the sake of reproducibility [1].

**Real data.** We consider two real datasets (other real datasets, with more than 1M edges, are available on the Jupyter notebook[1]):

- Openflights[2], a weighted graph of 3,097 nodes and 18,193 edges representing flights between the main airports of the world, the weight between two nodes corresponding to the number of daily round-trip flights between these airports;

- Wikipedia for Schools[3], a graph of 4,589 nodes and 106,644 edges representing the links between articles of Wikipedia selected for schools (see West et al. (2009)). The graph is considered as undirected (that is, there is an edge between two nodes if there is a link between the corresponding articles, in either direction). Weights are unitary.

Table 1 shows the results for the hierarchical clustering of these two graphs by Paris, the agglomerative algorithm based on linkage (10), and Newman's agglomerative algorithm based on the greedy maximization of modularity (Newman (2004)). The results are compared to a baseline obtained with a random agglomerative algorithm where node pairs are merged at random (among neighbor pairs). Since this algorithm returns a random hierarchy, we run it 100 times and give the average cost and quality of the hierarchy. Each metric is indicated for $\pi$ equal to $p$ (weighted metric, left columns) and $\pi$ uniform (unweighted metric, right columns). The best hierarchy according to each metric (highest value for the relative entropy, lowest value for Dasgupa's cost) is indicated in bold.

| Algorithm | RE | | DC | | | RE | | DC | |
|---|---|---|---|---|---|---|---|---|---|
| Paris | **2.77** | 2.91 | **0.167** | **0.130** | | **1.33** | 1.30 | **0.427** | 0.415 |
| Newman | 2.03 | **3.51** | 0.246 | 0.138 | | 0.951 | **1.33** | 0.461 | **0.410** |
| Random | 1.10 | 1.52 | 0.570 | 0.460 | | 0.762 | 0.811 | 0.645 | 0.630 |
| | (a) Openflights | | | | | (b) Wikipedia for Schools | | | |

Figure 1: Relative entropy (RE) and Dasgupta's cost (DC), both weighted (left columns) and unweighted (right columns), of the hierarchies of Openflights and Wikipedia for Schools returned by three algorithms: Paris, Newman, and random.

---

[1] `https://github.com/tbonald/hierarchy_metric`
[2] `https://openflights.org`
[3] `https://schools-wikipedia.org`

Not surprisingly, both Paris and Newman provide much better hierarchies than the random algorithm. Paris is the best algorithm for the weighted metrics, while Newman is the best algorithm for the unweighted metrics (except on Openflights, where Paris is better than Newman in terms of unweighted Dasgupta's cost). Considering the weighted metrics as more informative (just like the standard definition of modularity, given by (11) with $\pi = p$), both metrics tend to show that Paris provides a better hierarchy than Newman for both graphs.

**Synthetic data.** To differentiate both metrics, we proceed with the following experiments. We generate two noisy versions of the same graph, say $G_1$ and $G_2$, and return the corresponding hierarchies, say $H_1$ and $H_2$, by some hierarchical clustering algorithm. We then assess the ability of each metric to identify the best hierarchy associated to each graph (e.g., $H_1$ should be better than $H_2$ for graph $G_1$). The classification score (fraction of correct answers) is indicated for each metric and for two algorithms, Paris (in black) and Newman (in grey). Each classification score is based on 1000 samples of the graphs $G_1$ and $G_2$; the original graph $G$, generated at random, has 100 nodes and the two graphs $G_1$ and $G_2$ are derived from $G$ by replacing some fraction of the edges at random (i.e., one of both ends is chosen uniformly at random among the set of other nodes). The higher the distance between $G_1$ and $G_2$, the easier the classification task (since the corresponding hierarchies $H_1$ and $H_2$ differ more and more) and the higher the classification score.

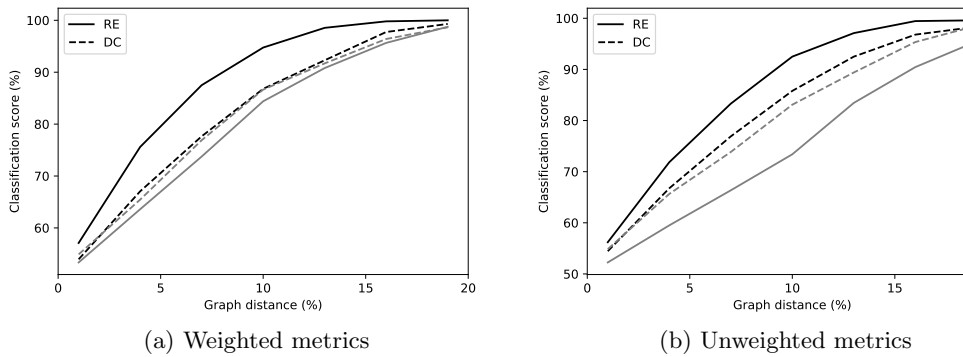

(a) Weighted metrics           (b) Unweighted metrics

Figure 2: Classification score of relative entropy (RE) and Dasgupta's cost (DC) with respect to the distance between the graphs (in fraction of randomly replaced edges), for Paris algorithm (black lines) and Newman algorithm (grey lines).

Both metrics tend to show that the Paris algorithm (in black) provides a better hierarchy than Newman's algorithm (in grey), the difference being more significant with the RE metric. For both the weighted and unweighted metrics, the best classification score is obtained by RE with the Paris algorithm. For a graph distance of 10%, the corresponding classification score is equal to 94.8% for the weighted metric and 92.5% for the unweighted metric, while it does not exceed 86.8% in all other cases.

## 7    Conclusion

We have proposed a novel metric for assessing the quality of hierarchical graph clustering, motivated by the problem of reconstruction of the graph from the dendrogram. This metric is the relative entropy between two probability distributions on the corresponding nested clustering, induced by edge sampling and independent node sampling, respectively. We have proved that, like Dasgupta's cost function, our quality metric is consistent in the sense that, if the original graph has a hierarchical structure, then the best hierarchy according to this metric is the underlying hierarchy of the graph. Experiments on both real and synthetic data tend to show that the relative entropy is both meaningful and significantly different from Dasgupa's cost. In future work, we would like to better characterize these differences, i.e., to identify the types of hierarchical structures that are best detected by each metric.

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
