# OpenReview forum: "Learning Graph Representations by Dendrograms"
_ICLR.cc/2019/Conference_

### Official Review · AnonReviewer3 · 2018-10-30
**Learning Graph Representations by Dendrograms**

**Rating:** 5
**Confidence:** 3

**Review:**

This paper suggests a new metric for assessing the quality of hierarchical clustering. Dasgupta recently suggested the first such metric with interesting properties. This has encouraged a number of recent works about designing algorithms that work well for the metric and other similar metrics. This paper suggests a new metric for evaluating hierarchical clustering of given graph data.

Here are the main comments about the paper:
- I am not convinced about the advantages of the new metric over the previously suggested metrics by Dasgupta and Cohen-Addad et al.
    - Theoretical analysis shows properties of the new metric that are similar to that of Dasgupta (since the metric itself has similarities). However, the advantage of the new metric is not very clear.
    - Experimental analysis just shows that the new metric is different from Dasgupta’s but again there is no evidence to suggest why the new metric may be better.

- In the abstract it is mentioned that “The best representation of the graph for this metric in turn yields a novel hierarchical clustering algorithm.” I do not understand this. Which novel algorithm is being referred to?

- Again, it is mentioned in the abstract that “Experiments on both real and synthetic data illustrate the efficiency of the approach”. What efficiency is being referred to here and what is the approach? What I see is that known clustering algorithms are used to compare the new metric with the previous one by Dasgupta.

Overall, I think more work is needed in this paper. There are some non-trivial observations but unless the authors make the motivation for defining this new metric for evaluation more clear.

Other comments:
- Section 5: NP-hardness has not been shown and it is just mentioned that the authors believe that the problem is NP-hard just as the problem associated with the cost function of Dasgupta et al.

---

> ### Public Comment · (anonymous) · 2018-11-23
> **An interpretable metric**
>
> As stated in our previous post, our main contribution is a metric that is *interpretable* in terms of graph reconstruction. We do not claim that our metric is better than existing ones. It is simply different. Its theoretical and practical interests come from the fact that it is derived from an optimization problem (minimizing the distance between the original graph and its reconstruction). There is no such interpretation with Dasgputa's cost.
>
> The algorithm mentioned in the abstract is the agglomerative algorithm based on linkage (10). This is a by-product of our approach. The main contribution is, again, the metric itself. The "efficiency of the approach" refers to the ability of our metric to detect "good" hierarchies (like Dasgupta's cost).
>
> We agree that the motivations of the paper need to be clarified in the abstract / introduction of the paper and we commit to do so if the paper is accepted.
>
> The authors

---

### Official Review · AnonReviewer1 · 2018-10-31
**Clearly written but the overall quality is not high enough.**

**Rating:** 5
**Confidence:** 4

**Review:**

This paper proposes a new formulation of hierarchical clustering for graphs.

Quality:
The proposed formulation has not been analyzed in detail and its advantage is not clear compared to existing approaches.
In addition, there are some existing measures for hierarchical clustering, for example, the dendrogram purity[1].
It would be interesting to analyze the relationship between the proposed method and such existing measures.
[1] Heller, K. A. and Ghahramani, Z.: Bayesian Hierarchical Clustering, ICML2005

Clarity:
This paper is clearly written and easy to read.
The proposed criteria is carefully derived and well explained.
I feel that the title of the paper is not appropriate as it says the paper is about graph representation learning while a graph (representation) is already given as an input in the setting discussed in this paper. "Learning hierarchical representation ..." would be better.

Originality:
The originality is not high as most of theoretical discussion is based on the existing work and the resulting hierarchical clustering algorithm is a straightforward extension of the average linkage method.
Of course, it is quite interesting if a minor change makes a big difference in clustering performance (theoretically and/or empirically), but such result is not given.

Significance:
Significance of the contribution is not high as the advantage of the proposed formulation is not clear.
One of interesting questions is: how about higher order relationships between nodes?
The proposed method takes up to second order relationships between nodes, that is, edges into account.
Since the proposed formulation can naturally include higher order relationships, it would be interesting to analyze such relationships in hierarchical clustering.

Pros:
- The paper is clearly written.
- The proposed formulation of hierarchical clustering is interesting.
Cons:
- The advantage of the proposed formulation is not presented.
- Experiments are not thorough.

---

> ### Public Comment · (anonymous) · 2018-11-23
> **Quality, originality and significance**
>
> We thank the reviewer for her/his comments.
>
> Quality:
> * To our knowledge, the only existing metric is Dasgupta's cost; the dendrogram purity applies to labelled data only while we don't have any label in our setting.
> * The paper includes some experiments comparing both metrics (Dasgupta's cost and our metric) qualitatively; the objective is not to "prove" that one is better than the other, as this would depend on the considered task in any case (like for most performance metrics in the setting of unsupervised learning).
> * The key point is that our metric is *interpretable* in terms of optimal graph reconstruction (unlike Dasgupta's cost). This is interesting from both theoretical and practical viewpoints.
>
> Originality:
> * Our metric is the first to be interpretable in terms of graph reconstruction. A hierarchy is "better" than another in the sense of our metric if it allows to better recover the initial graph.
>
> Significance:
> * We agree that higher-order relationships can be naturally be included to our metric (showing again its interest!), and we would mention that explicitly in the paper if it is accepted (thanks for the suggestion :-). However, we do think that the proposed metric, based on pairwise relations, is both relevant and easier to interpret.
>
> In short, the main advantage of our metric is that it is *interpretable*, which makes it easy to understand, to apply, and to enrich.
>
> The authors

---

### Official Review · AnonReviewer2 · 2018-11-04
**Relative entropy for evaluating hierarchical clustering**

**Rating:** 4
**Confidence:** 4

**Review:**

This paper introduces a relative entropy metric for evaluating hierarchical clustering. The paper is not well-written and its contributions are not clear to me. The abstracts and introduction promising for a novel hierarchical clustering algorithm, but I can only understand that they are using an agglomerative algorithm with the inter-cluster similarity of Eq. 10.
They show that their similarity metric reduces to the previous work by setting \pi = p or uniform. However, in the experiments, they only use these two cases, which does not support the importance of using relative entropy metric.  I guess picking the best \pi is an important part of this approach which has been left out.
The authors violate the blindness of the paper by including a link to their github page, for which an anonymous repository should have been used.
It also worth noting that nowadays the graph representation term is often used for graph embedding, which makes the title very misleading.

---

> ### Public Comment · (anonymous) · 2018-11-23
> **A novel metric for hierarchical graph clustering**
>
> We thank the reviewer for her/his comments. The main contribution of the paper is a novel metric for evaluating the quality of any hierarchical clustering. It is independent of the clustering algorithm, unlike what is suggested in the review. We agree that the title of the paper is misleading in this respect, and we would change it in case of acceptance (the title of the reviewer's comment is a good candidate :-).
>
> The literature on metrics for hierarchical graph clustering is very limited and we claim that ours, based on a totally new approach and finding roots in information theory, may be very interesting for researchers and engineers working on graph data. The reviewer is right in saying that graph representation generally refers to graph embedding, but it is not the only one. It is the role of top conferences like ICLR to present fresh, disruptive ideas to the community.
>
> The authors

---

### Meta-Review · Area_Chair1 · 2018-12-14
**Meta-Review for Learning Graph Representations paper**

**Confidence:** 5
**Recommendation:** Reject

**Metareview:**

All reviewers agree to reject. While there were many positive points to this work, reviewers believed that it was not yet ready for acceptance.